# Handcrafting Generative Adversarial Networks for Fashion Design Generation

**Abstract.** Natural images, particularly, those related to fashion exhibit structural coherence including symmetry. This is also true for natural textures which comprise of a repeating periodic pattern. In this paper, we build on existing work and demonstrate that it is possible to generate images of desired characteristics. We model losses based on symmetry and periodicity to train Generative Adversarial Networks. We extend the notion of a fictional generator that modifies the generated images to those with expected properties. The discriminator is trained to classify them equivalently as *real* or *fake*. We also propose a regularizer on the generator to produce images with specified properties. We use data from fashion catalog to train the models and show that generating high resolution images is feasible. We present numerical results to quantitatively evaluate the methods in comparison to existing ones and include images to qualitatively show they produce images of superior quality.

**Keywords:** Generative Adversarial Networks, Regularization, Symmetry, Periodicity, Tiling, Generative Models, Generator, Discriminator

## 1   Introduction

Generative models are being developed for several applications. They typically learn the distribution of the data and can be used to produce new ones that look realistic. Generative Adversarial Networks (GAN) [3] are a class of generative learning algorithms. In a GAN, Generator and Discriminator networks compete to create and distinguish data respectively in a actor-critic manner. However, they suffer from several drawbacks including mode collapse and are notorious to latch onto only a few modes. They also produce distorted images with missing details. GANs are generally tricky to train, as perturbations in the hyperparameters (dimensionality, learning rate, etc.) result in substantially varying outputs.

We conducted experiments with DCGAN [9], a commonly used GAN framework to train for fashion images. Several tricks have been proposed to make it easier to train GANs [2], [11]. Regularizers have also been developed to overcome some of the challenges [10], [6]. However, despite these improvements, the images generated suffer from lack of detail and distortions. Hence we attempt to handcraft GANs to produce images with features that are expected.

There are some recent efforts in this direction [7]. The method in [8] controls the generated images by modifying the architecture of the generator. The

methods in [13, 12, 4] propose solutions to handle human models wearing the items.

Fashion images exhibit a high amount of symmetry and the patterns are periodic. We try to teach a GAN to learn them, thereby minimizing the artifacts. We extend the idea of a virtual generator [7] that transforms a generated image to the one with specified properties to other related problems. We run these images through the discriminator and train it to classify them equivalently as *real* or *fake*. We observed that the images thus generated adhere better to the expected characteristics and suffer less from aberrations. To overcome issues of stability and mode collapse, we use a regularizer on the discriminator [10]. We extend the work to generate images of periodic patterns that can be tiled.

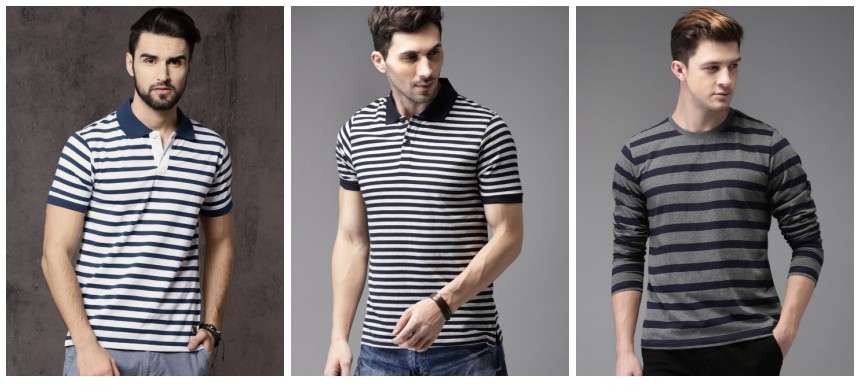

Fig. 1: Example stripe t-shirts.

Stripe patterns will comprise of multiple straight lines of various colors. Hence when a GAN is trained on them, it is expected to retain these properties. However, since the input data (Fig. 1) has significant variations, the generated stripes also suffer from them. Thus there is a need for a solution to minimize them and produce stripe patterns that resemble the original ones. We propose a solution by exploiting the inherent periodicity of them and add a regularizer to ensure that the generated stripes are straighter.

We evaluate the results quantitatively and qualitatively to demonstrate the merits of the proposed solutions. We compare them with previous methods for the purpose of fashion design generation.

## 2 Background

We review the tricks that are commonly used to train GANs and a recent regularizer that is proposed to prevent issues experienced during training.

## 2.1 Regularizer

GANs are generally hard to train. The optimization algorithm to update the weights of the Generator and the Discriminator might lead to local optimums instead of the global one. A recent regularizer proposed in [10] plays an effective role in regularizing the Jenson-Shannon divergence. By adding a regularizer based on gradients, the authors achieve a more stable GAN training procedure that enabled them to train convergently for a longer period of time. The loss function is defined as:

$$F_\gamma(P, Q; \varphi) = \mathbb{E}_P[ln(\varphi)] + \mathbb{E}_Q[ln(1 - \varphi)] - \frac{\gamma}{2}\Omega_{JS}(P, Q; \varphi)$$

$$\Omega_{JS}(P, Q; \varphi) = \mathbb{E}_P[(1 - \varphi(x))^2||\nabla\phi(x)||^2] + \mathbb{E}_Q[(\varphi(x))^2||\nabla\phi(x)||^2]$$

where $P$ and $Q$ refer to the True and Generated Data Distribution respectively. $\phi = \sigma^{-1}(\varphi)$ denotes logits of the discriminator $\varphi$. $\Omega_{JS}$ is the regularization term weighted by $\gamma$.

## 2.2 Symmetry

Fashion images exhibit a great amount of symmetry. It was recently demonstrated [7] that it is possible to learn the distributions of the original as well as the mirrored versions of them. Additional losses have been added wherein the training and generated images and their mirrored versions are equivalently classified as *real* or *fake* by the discriminator. The notion of a virtual generator which produces images of desired properties has been introduced.

The results in [7] were reported using DCGAN but we realized that issues such as stability and generalization remain. To overcome them, we use the regularizer [10] described in the previous section. We extend the work to train for stripe tshirts and check shirts where the improvements due to symmetry are more pronounced (Fig. 4 and Fig. 5). We add additional layers to generate images of higher resolution and demonstrate that it is possible to generate images of good quality (Fig. 6).

$$\min_G \max_D V(G, D) = \mathbb{E}_{x \sim p_d(x)}[log(D(x))] + \mathbb{E}_{z \sim p_z(z)}[log(1 - D(G(z)]$$

$$\min_G \max_D V(G, D) = \mathbb{E}_{x \sim p_d(x)}[log(D(x))] + \mathbb{E}_{z \sim p_z(z)}[log(1 - D(G(z)))]$$

$$+ \mathbb{E}_{x \sim p_{fd}(x)}[log(D(x))] + \mathbb{E}_{z \sim p_z(z)}[log(1 - D(G'(z)))]$$

where $p_d$ is the distribution of the data, $p_{fd}$ is the distribution of the flipped images, $p_z$ is the latent distribution (typically a MVN), $G'(z)$ is the horizontal mirror image of $G(z)$.

# 3  Tiling

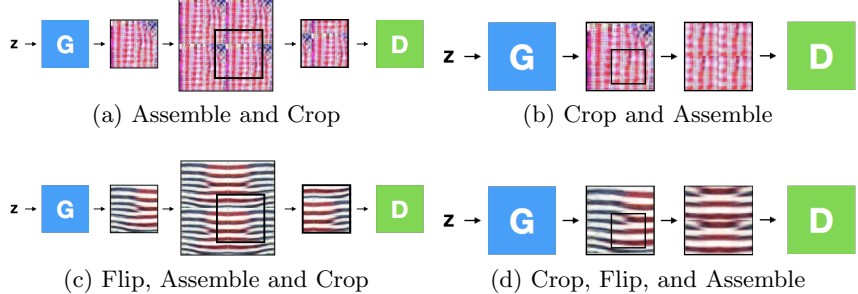

(a) Assemble and Crop          (b) Crop and Assemble

(c) Flip, Assemble and Crop      (d) Crop, Flip, and Assemble

Fig. 2: Schematics for Tiling Schemes

The ideas that are presented in [7] for symmetry can be extended to train for other properties as well. Generally, textures are periodic and swatches of them should be tileable without discontinuities at the merging boundaries [1]. We present some ways by which this can be achieved. We create a fictional generator that would produce a tiled image from the generated image. There two choices exist - (a) Assemble and Crop (b) Crop and Assemble. It is generally expected that strategy (a) will be better than (b) since the Generator may produce images where the boundaries can be better merged. But (b) will ensure that the generated image itself is a tiled version and thereby leading to better tileable images. It is nevertheless good to test the hypothesis as to whether the generator can handle the discontinuities and produce images that do not suffer from them. Unlike in [7], we do not calculate the synthesized losses for the original images (training data) used for training since they would induce artefacts.

## 3.1  Assemble and Crop

We present additional work to overcome the scalability issue when designing textured patterns using GANs (i.e., assembling the images together to fill a given silhouette). A checkered or stripe pattern of a fashion item has a fixed repeat pattern size and should be repeatable. However, when a Generator learns the pattern of a swatch, it may not be possible to tile it repeatedly due to inconsistencies in the boundaries between them. To overcome this, we propose tiling the generated image four times and randomly crop a swatch of the original size from it (Fig. 2). The discriminator should classify it equivalently (*real* or *fake*) as the generated image.

## 3.2   Crop and Assemble

We crop a random region within the generated image and assemble it to fill the original size (Fig. 2). This method will ensure that there is better coherence in the generated image since it is a tiled version itself. The generator, if trained well, is expected to overcome the discontinuities at the boundaries.

## 4   Symmetry and Tiling

There are certain textures that obey symmetry as well (e.g., stripes and some check patterns). Hence we propose to assemble a generated image and it's mirrored version to produce an image which satisfies both symmetry and periodicity properties (Fig. 2). We take a reflection of the generated image and combine it with the original to form a sub-image. This sub-image is replicated to generate an image of the original size. We experiment with this hybrid strategy for both Assemble & Crop and Crop & Assemble methods.

## 5   Regularizer to Produce Desired Patterns

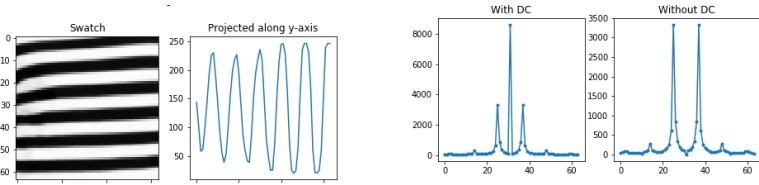

Fig. 3: Stripe pattern, row projection and spectrogram.

We present an approach to train GANs to generate better stripe patterns. Figure 3 shows an example stripe pattern cut from a catalogue image. It can easily noticed that that it suffers from several artefacts compared to the original cloth pattern of it. Hence it is expected that a GAN trained on such images will produce similar ones. There is an inherent periodicity in the stripe pattern and if exploited well could lead to better generated ones.

We convert the image from RGB to gray scale and process it. In this example, the stripe pattern is horizontal and hence the row projection of it is expected to be periodical. From the figure, it can be noticed that it is roughly periodic owing to the variations in the original stripe. Hence the spectrogram of it is expected to have a predominant peak. After removing the DC component, it can be noticed that there is a clear peak for the dominant frequency. We propose to add a constraint on the peak in the spectrogram of the generated images.

Ideally, there should be a single predominant peak but it may not always hold owing to the distortions introduced due to the way the product is worn by

a person. Hence we propose to maximize the peak value in the spectrogram and add a regularizer for it in the loss functions used to train GAN.

We add a regularizer term $G_{fft}$ to check if the peak in the spectrogram is predominant:

$$G_{fft} = 1 - \frac{||fraction\_spectrum||_2^2}{||total\_ac\_spectrum||_2^2} \qquad (1)$$

where $||fraction\_spectrum||_2^2$ represents energy of the 3-point band around the peak in the spectrogram. $||total\_ac\_spectrum||_2^2$ is the total energy of the AC component.

# 6    Performance Evaluation

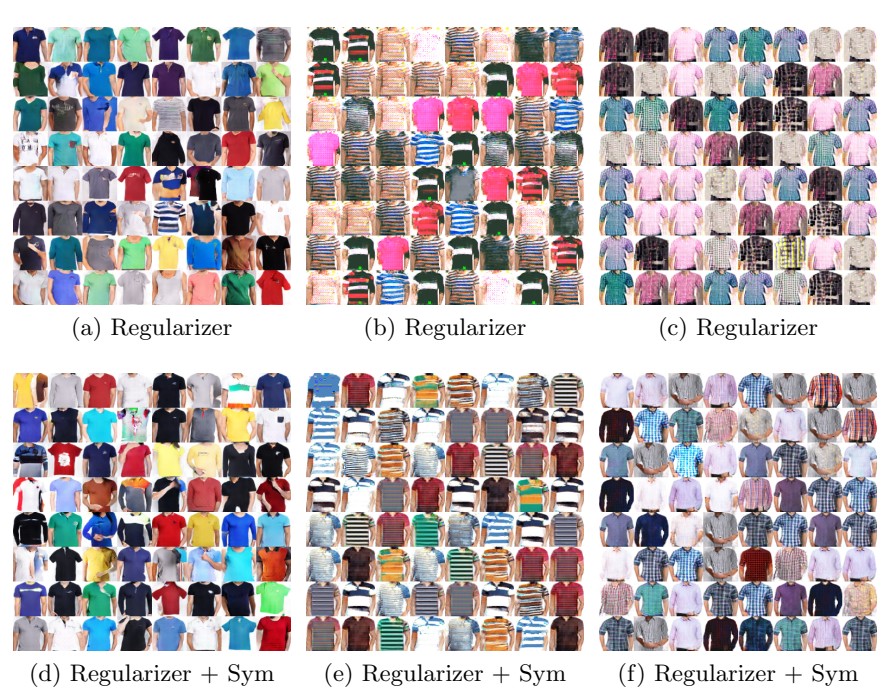

(a) Regularizer          (b) Regularizer          (c) Regularizer

(d) Regularizer + Sym     (e) Regularizer + Sym     (f) Regularizer + Sym

Fig. 4: Full body generated images for different patterns. (a, d) for Solids, (b, e) for Stripes and (c, f) for Checks.

We train the proposed methods on fashion catalog images by modifying the code of [5]. The images used to train the methods comprise of numerous variations available on a fashion e-commerce platform (Table 1). We qualitatively evaluate the performance of the methods by comparing our results with those from earlier schemes. We show images from the later epochs of the training.

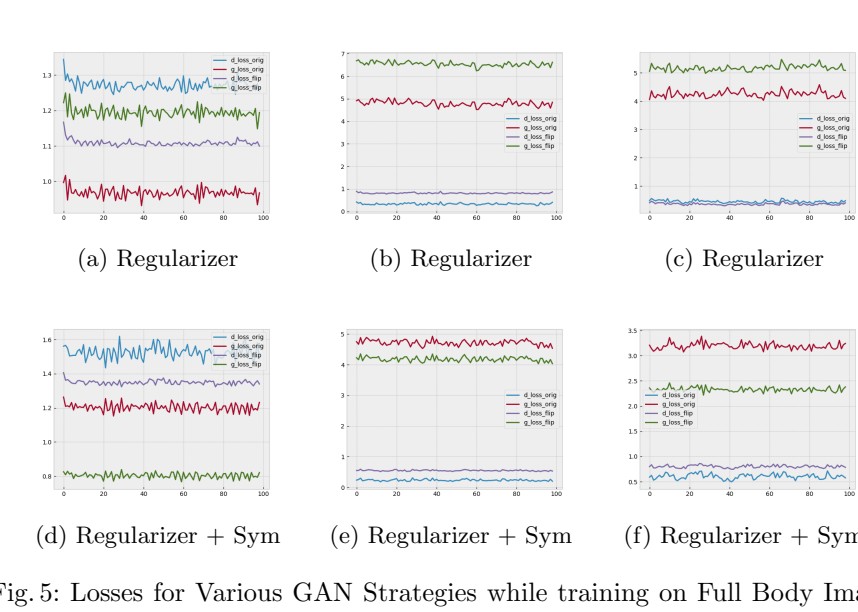

(a) Regularizer          (b) Regularizer          (c) Regularizer

(d) Regularizer + Sym    (e) Regularizer + Sym    (f) Regularizer + Sym

Fig. 5: Losses for Various GAN Strategies while training on Full Body Images. (a, d) for Solids, (b, e) for Stripes and (c, f) for Checks.

| Article/Pattern | No. of images |
|---|---|
| Checks Shirts | 40426 |
| Stripes T-Shirts | 41886 |
| Solids T-Shirts | 42225 |

Table 1: Size of the training data set

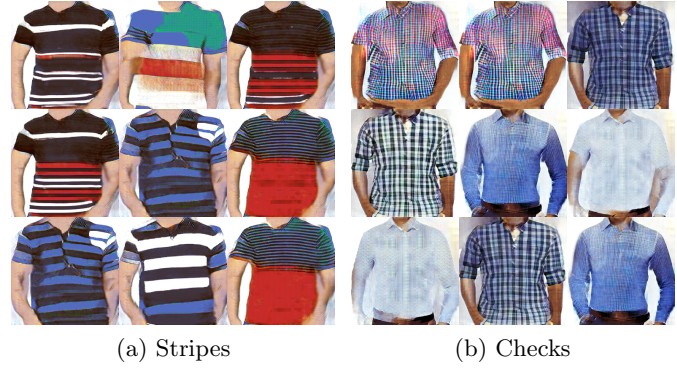

(a) Stripes          (b) Checks

Fig. 6: The Regularizer + Sym strategy produces images of resolution 256x256 quite appealingly for various design patterns, as they have more detail in them.

We build on the previous work in [7] by using regularizers [10] to improve training of GANs [11]. We add additional layers to the DCGAN framework to generate images of higher resolution. We demonstrate the efficacy of the methods by comparing the loss plots for the images synthesized from the generated images with the original ones. The generated images from DCGAN trained using the regularizers and the proposed methods are compared visually. The results reported are for images of resolution 64x64 unless otherwise stated.

The plots in Fig. 5 (Sec. 2) show that the mirrored images of synthesized images are not within the scope of the distribution learnt by the Generator trained using DCGAN and regularizer. While the regularizer reduces the losses and improves the stability of training, the losses are still higher. The addition of the symmetry constraint has reduced the losses for both the generated images and their mirrored ones while also bringing them closer. The visual quality of the generated images is also superior and they suffer less from artefacts (Fig. 4). The higher resolution images have greater detail (Fig. 6).

We show the results for tiling (sections 3 and 4). with the four strategies for stripe and check patterns (Fig. 7). It can be noticed that the images thus generated are more coherant and display better structure. They compare well with those generated using the other schemes. The generator is able to handle the discontinuities at the boundaries to a great extent.

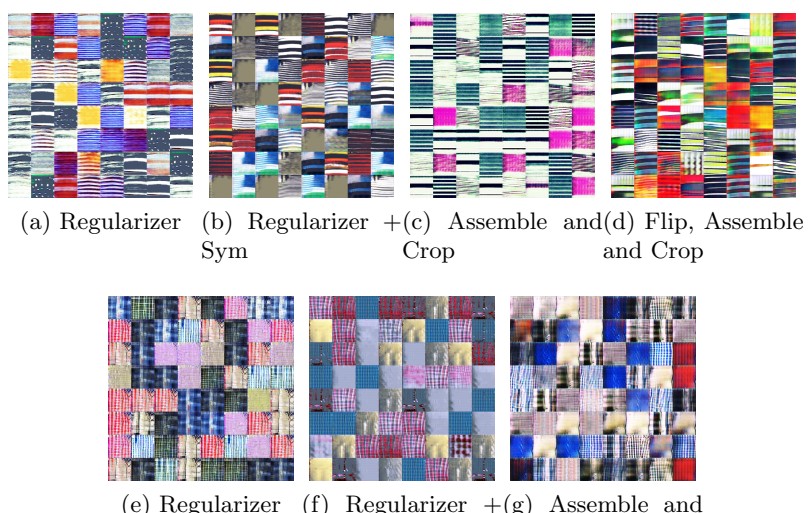

(a) Regularizer    (b) Regularizer +(c) Assemble  and(d) Flip, Assemble
                   Sym            Crop              and Crop

(e) Regularizer    (f) Regularizer +(g) Assemble and
                   Sym            Crop

Fig. 7: Generated swatches of different patterns. (a, b, c, d) are Stripes and (e, f, g) are Checks.

Figure 8 shows the results for the method for stripe pattern generation (Sec. 5). The results using [10] are given in Fig. 8g, 8h and 8i. When the symmetry losses from [7] are used in addition to [10], the resultant images are Fig. 8d, 8e, and 8f. In addition to them, when we add the regularizer for peak in spectrogram (equation 1), we obtain those in Fig. 8a, 8b and 8c. From these results, it can be easily perceived that the proposed peak regularizer has resulted in straighter stripes with superior quality.

## 7    Conclusions

We introduced methods by which GANs can be trained to generate images of desired characteristics. In particular, we improve earlier work on symmetry by adding a regularizer and add additional layers to generate images of higher resolution. We show that the losses for images synthesized from the generated images are comparable to those of the generated ones and that the generated images are of better quality. We extended the methods to tile swatches of checks and stripes and show that the proposed methods work well for other applications as well. We also introduced a regularizer to generate stripe patterns that are straighter and cleaner. We demonstrated the effectiveness of the methods by showing performance improvement over existing schemes.

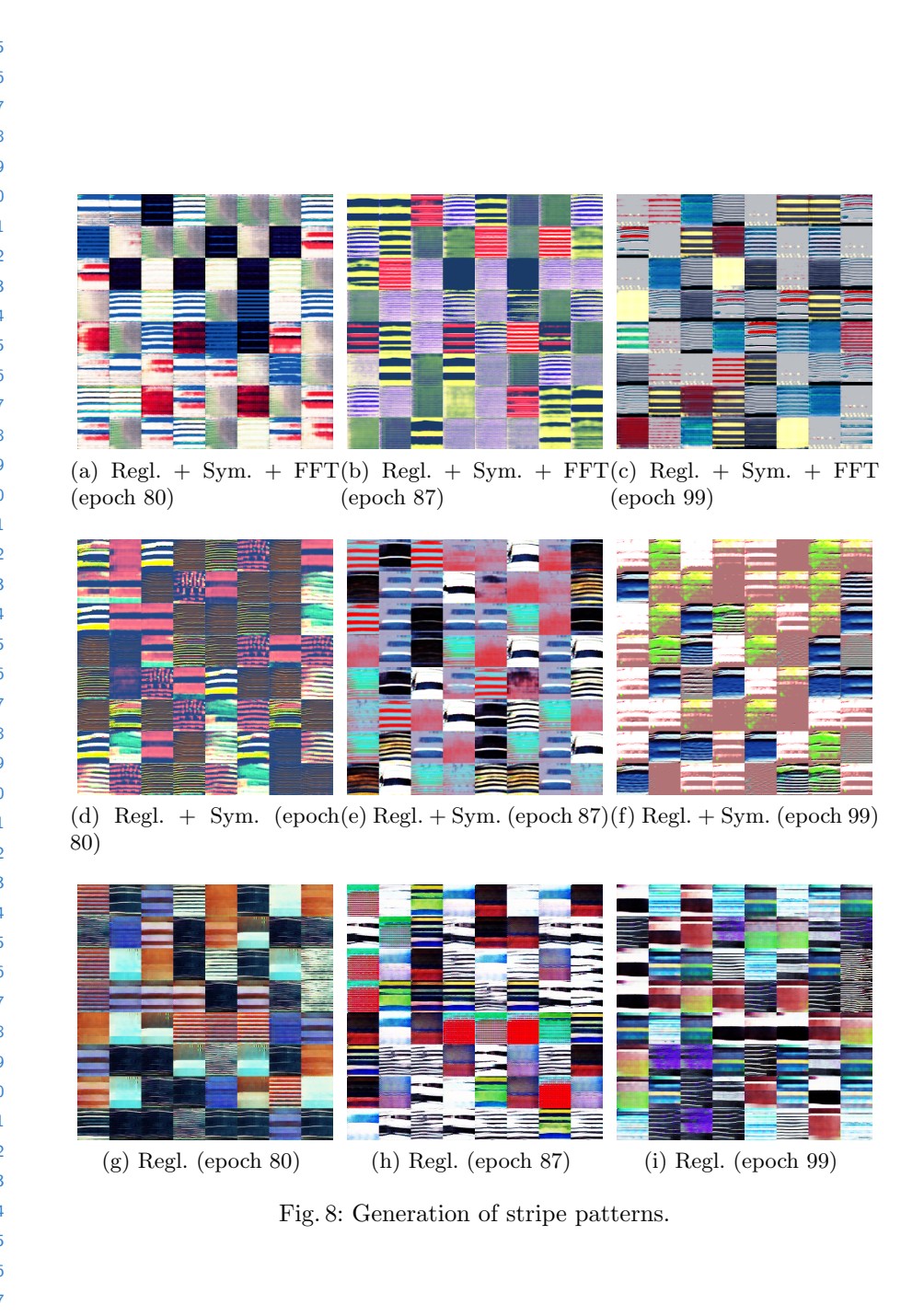

(a) Regl. + Sym. + FFT (b) Regl. + Sym. + FFT (c) Regl. + Sym. + FFT (epoch 80)                (epoch 87)                (epoch 99)

(d) Regl. + Sym. (epoch (e) Regl. + Sym. (epoch 87) (f) Regl. + Sym. (epoch 99) 80)

(g) Regl. (epoch 80)          (h) Regl. (epoch 87)          (i) Regl. (epoch 99)

Fig. 8: Generation of stripe patterns.

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
