# OpenReview forum: "Handcrafting Generative Adversarial Networks for Fashion Design Generation"
_thecvf.com/ECCV/2020/Workshop/VIPriors — Submitted to VIPriors_

### Official Review · AnonReviewer1 · 2020-07-21
**Good idea, but execution needs work**

**Confidence:** 4
**Rating:** 6

**Review:**

[Summary] In 2-3 sentences, describe the key ideas, experiments, and their significance.

The authors propose to modify GANs by augmenting the generator outputs according to visual priors on their fashion dataset. The results are visually more appealing.

[Strengths] What are the strengths of the paper? Clearly explain why these aspects of the paper are valuable.

Strong idea; topical submission.

[Weaknesses] What are the weaknesses of the paper? Clearly explain why these aspects of the paper are weak.

- Presentation needs a lot of work (see detailed comments);
- Incremental contribution over existing works;
- Lack of quantitative evaluation.

[Overall rating] Paper rating: marginal accept

[Justification] The idea is very interesting and applicable to the workshop. Please take the time to revise the final version of your paper using the detailed comments.

[Detailed comments] Additional comments regarding the paper (e.g. typos or other possible improvements you would like to see for the camera-ready version of the paper, if any.)

- Section headers should be revised to match paper conventions (i.e. Related Works, Method, Experiments, Discussion)
- Equations in section 2 either need more explanation or need to be replaced by references
- Sections 1 and 2 contain very broad and imprecise phrasing, e.g. "The notion of a virtual generator which produces images of desired properties has been introduced", "[...] transforms a generated image to the one with specified properties to other related problems."
- Figures should be close to their place in the text

Minor comments:

- Vague phrasing: lines 48 "them"
- Line 227: where is this term added?
- Line 333: "coherant"

---

### Official Review · AnonReviewer2 · 2020-07-28
**Handcrafting Generative Adversarial Networks for Fashion Design Generation**

**Confidence:** 4
**Rating:** 3

**Review:**

1. [Summary] In 2-3 sentences, describe the key ideas, experiments, and their significance.

 This paper describes a method based on GANs to generate fashion images with different textures.

2. [Strengths] What are the strengths of the paper? Clearly explain why these aspects of the paper are valuable.


3. [Weaknesses] What are the weaknesses of the paper? Clearly explain why these aspects of the paper are weak.

 -	The structure of the paper is not clear.
 -	Fashion related works exist. Authors do not mention anything about it. Only background for GANs.
 -	Other topics where GANs are used similarly?
 -	It lacks research motivation.

4. [Overall rating] Paper rating.

 3

5. [Justification of rating] Please explain how the strengths and weaknesses aforementioned were weighed in for the rating.

 Weaknesses from point 3 justify rating

6. [Detailed comments] Additional comments regarding the paper (e.g. typos or other possible improvements you would like to see for the camera-ready version of the paper, if any.)

---

### Decision · Program_Chairs · 2020-07-29

**Decision:**

Reject

**Comment:**

After considering the reviews and further discussion we concluded that although idea seems interesting, the paper was hard to read and the related works section is weak. We therefore believe that the paper needs one more iteration before it could be accepted.